

# Empirical evaluation of methods for *de novo* genome assembly

Firaol Dida and Gangman Yi

Department of Multimedia Engineering, Dongguk University, Seoul, South Korea

## ABSTRACT

Technologies for next-generation sequencing (NGS) have stimulated an exponential rise in high-throughput sequencing projects and resulted in the development of new read-assembly algorithms. A drastic reduction in the costs of generating short reads on the genomes of new organisms is attributable to recent advances in NGS technologies such as Ion Torrent, Illumina, and PacBio. Genome research has led to the creation of high-quality reference genomes for several organisms, and *de novo* assembly is a key initiative that has facilitated gene discovery and other studies. More powerful analytical algorithms are needed to work on the increasing amount of sequence data. We make a thorough comparison of the *de novo* assembly algorithms to allow new users to clearly understand the assembly algorithms: overlap-layout-consensus and de-Bruijn-graph, string-graph based assembly, and hybrid approach. We also address the computational efficacy of each algorithm's performance, challenges faced by the assembly tools used, and the impact of repeats. Our results compare the relative performance of the different assemblers and other related assembly differences with and without the reference genome. We hope that this analysis will contribute to further the application of *de novo* sequences and help the future growth of assembly algorithms.

Corresponding author
Gangman Yi, gangman@mme.dongguk.edu

## INTRODUCTION

Analyzing DNA sequences has become a critical aspect of basic biological research in a variety of applied fields, such as medical diagnosis, biotechnology, forensic biology, virology, and biological systematics. The identification of diseases like various cancers is possible via comparisons of stable, mutated DNA sequences (*Chmielecki & Meyerson, 2014*) and can be used as a guideline for patient treatment (*Pekin et al., 2011*). Personalized medical attention can be provided via a swift approach to DNA sequencing and by recognizing and listing more organisms (*Abate et al., 2013*).

*De novo* assembly (*Miller, Koren & Sutton, 2010*; *Nagarajan & Pop, 2013*; *Denton et al., 2014*) refers to the sequencing of a novel genome where no reference sequence for alignment is available. Sequence reads are assembled as contigs, and data coverage quality of *De novo* assembly depends on the size and continuity of the contigs (*Park, 2017*; *Nagarajan & Pop, 2013*). Precise genome reconstruction is imperative, as the consistency and the base accuracy of the assembly will influence the outcomes of all downstream analyses (*Denton et al., 2014*). The assembly problem becomes more complicated and

computationally intensive (*Head et al., 2014*) with increasing efforts to sequence and assemble the genomes of more species, particularly with short, inaccurate sequence reads and genomic repeats (*Liao et al., 2019*; *El-Metwally, Zakaria & Hamza, 2016*). Compared to conventional approaches, such as Sanger sequencing (*Beck, Mullikin & lesb@ mail. nih. gov NISC Comparative Sequencing Pro-gram Biesecker Leslie G, 2016*; *Grada & Weinbrecht, 2013*), next-generation sequencing(NGS) enables quicker (*Khodakov, Wang & Zhang, 2016*; *Metzker, 2010*), more precise characterization of any species (*Mestan et al., 2011*).

The expeditious sequencing achieved using modern DNA sequencing technology was instrumental at sequencing a complete DNA sequence or genomes of different types and species, including humans, organisms, plants, and microbes (*Vega, 2019*). Obtaining genome sequences is now much simpler and cheaper than it was during the Human Genome Project (*Collins, Morgan & Patrinos, 2003*; *Mardis, 2011*), thanks to modern methods that have been developed over the last two decades.

In genome research, *de novo* genome assembly is a fundamental endeavor that has led to the creation of high-quality reference genomes (*Goffeau et al., 1996*; *Myers et al., 2000*; *Bonfield, Smith & Staden, 1995*) for many haploid or highly inbred species and has facilitated gene discovery, comparative genomics, and other studies (*Chin et al., 2016*). Increasingly powerful analysis algorithms are needed to keep pace with the rising availability of sequence data. This is of particular significance as large genomes are assembled, where datasets can be up to hundreds of gigabytes in size (*Simpson & Durbin, 2012*). As *de novo* assembly usually allows queries to be performed over the entire set of sequence reads, vast datasets present a practical problem for assembly software developers and users. At present, a single computer with a large memory, usually hundreds of gigabytes (*Li et al., 2010*; *Gnerre et al., 2011*) or a large distributed cluster of connected computers (*Simpson et al., 2009*; *Boisvert, Laviolette & Corbeil, 2010*) is required for an assembler (*Simpson & Durbin, 2012*).

Among the advantages of *de novo* assembly is that it can produce precise reference sequences even for sophisticated or polyploid generations, provide valuable information for mapping novel organism genomes or completing genomes of known organisms, resolve immensely similar or repetitive regions for accurate *de novo* assembly (*Nguyen et al., 2018*), and recognize structural variants and complex rearrangements.

Sequencing the entire genome remains a challenging task. One of the most critical and challenging problems in bioinformatics is the sequence assembly problem. The purpose of genome assembly is to recreate a full genome from several relatively short sequences. Overlaps can be joined to form contigs in reads from the same area of the genome, but genomic repeats longer than overlaps cause obscure reconstruction and fragmentation of the assembly (*Phillippy, Schatz & Pop, 2008*; *Nagarajan & Pop, 2009*). Most genomes, particularly eukaryotic genomes, are highly repetitive and complicate the assembly by obscuring the reads' interrelationships with many false options. There are two strategies to tackle this fundamental constraint: increasing the effective read length and separating non-exact repeats based on copy-specific variants (*Koren et al., 2017*).

In repetitive genome regions, it is difficult to accurately assemble short reads, so imprecise or unsolved assemblies may be generated. The repetition of the genome regions has been

enhanced using long-read single molecule sequences (SMS) technologies such as Pacific Biosciences and Oxford Nanopore (*Reuter, Spacek & Snyder, 2015*). However, several long stretches of repetitive DNA do not yield to these approaches.

With the recent development of long-read technology, having near-finished assemblies became possible. However, extracting information in long reads is still susceptible to errors, as repeats must be consistently overcome (*Myers Jr, 2016*). Attempts to overcome repeats that are essentially unresolvable from the readings at hand will lead to incorrect assemblies and eventually affect downstream scientific analysis: a procedure that can be motivated by the promise of a higher N50 score. However, given the data, a conservative approach that breaks the assembly at points of evident uncertainty may not yield the longest contigs that can be constructed (*Kamath et al., 2017*). In this sense, an assembler capable of recognizing and resolving all, and only those repeat patterns that are resolvable provided the read data available, should be an optimal assembler.

Genome assembly has attracted increased interest with the advent of NGS technologies (*Kim, Ji & Yi, 2020*; *Oxford Nanopore, 2020*; *PacBio, 2020*; *Illumina, 2020*; *DeciBio, 2020*; *Biosciences, 2020*; *DNALink, 2020*). While several genome assemblers are presented, *de novo* genome assembly using next- generation reads still faces four key challenges. The first challenge is sequencing errors, which contribute to to artifacts being included in the results of the assembly. Sequencing errors typically lead to a complex de Bruijn graph. The final results from a complex de Bruijn graph are generally unsatisfactory (*Liao et al., 2019*). The second challenge is the sequencing bias (*Rodrigue et al., 2009*). For instance, the base composition bias (favoring GC-balanced regions) of the Illumina sequencing platform typically results in an unequal depth of sequencing across the genome (*Sims et al., 2014*). The third challenge is the topological complexity of repetitive regions in the genome (*Liao et al., 2019*). Most genomes, particularly mammalian genomes, have some repetitions, which represent around 25–50% of the entire genome (*Kazazian, 2004*). The repeats create not only misconceptions or discrepancies in the results of the assembly but also irreconcilable sequence data depth. The fourth challenge is the use of large amounts of computing resources. Despite taking just a few minutes to assemble small genomes, such as bacterial genomes, it usually takes several days or even weeks to assemble large genomes, such as mammalian genomes.

This research aims to evaluate *de novo* assembly programs as a whole, to examine various aspects of assemblies thoroughly. It aims to see how the most recent genome assemblers performed on a sample of large-scale next-generation sequencing projects. The study is designed in response to the increasing use of NGS for *de novo* assembly and a number of genome assemblers. It aims to address questions like: How much do read types influence assembler performance? Which assembler is the most efficient?

The answer to these questions, as demonstrated below, *de novo* assembly depends very critically on the features of the genome, the nature of the sequencing experiments and the assemblers. Our results specify the complete technique that we used with each assembler for assembling each genome. All procedures and parameters are defined and the complete datasets used for each assembly are given in this paper. This should allow the replication

of some of our results along with the use of open-source assemblers. All data used in our assessments are also real sequence data from high-performance sequencing machines.

## *DE NOVO* ASSEMBLY METHOD

The rapid progress speed in sequencing technology has laid a solid foundation for the entire method of genome shotgun assembly (*He et al., 2013*; *Li et al., 2010*; *Ansorge, 2009*; *Fox, Filichkin & Mockler, 2009*; *MacLean, Jones & Studholme, 2009*; *Hall, 2007*; *Mardis, 2008*; *Metzker, 2010*; *Morozova & Marra, 2008*; *Shendure et al., 2004*). Some assemblers use methods that mainly deal with the sequence from the perspective of graphs. They can then use graph theory and algorithms to solve the assembly problem (*He et al., 2013*; *Wajid & Serpedin, 2012*). Other assemblers have been designed to follow Seed Extension (SE) methods (*Ahmed, Bertels & Al-Ars, 2016*; *Dohm et al., 2007*; *Jeck et al., 2007*; *Warren et al., 2007*; *Chu et al., 2013*). These types of methods primarily take advantage of knowledge on various insert sizes of Paired-end (PE) readings (*Batzoglou et al., 2002*; *Li et al., 2010*; *Boetzer et al., 2011*; *Huson, Reinert & Myers, 2002*; *Zerbino et al., 2009*; *Dayarian, Michael & Sengupta, 2010*). To some degree, these two types of approaches are capable of overcoming the difficulties associated with genomic repetition and non-uniform coverage. We will only incorporate graph-based approaches for this survey study.

### Overlap Layout Consensus (OLC) method

The OLC approach is based on graph overlaps. Fundamentally, this approach operates in three stages. First, overlaps (O) are found between all the reads. The OLC method then creates the layout (L) of all reads and overlaps the details in the graph. The consensus sequence (C) is then concluded from the multiple sequence alignments(MSA) (*Wang & Jiang, 1994*; *Idury & Waterman, 1995*).

An overlap graph represents the sequencing reads and their overlaps are used in the OLC process (*Miller, Koren & Sutton, 2010*; *Koren et al., 2012*). Overlaps must be pre-computed, and overlap detection between each pair is explicit, usually by all-against-all pairwise alignment (*Altschul et al., 1990*; *Haque, Aravind & Reddy, 2009*; *Giegerich & Wheeler, 1996*). The overlap graph indicates overlaps between reads with nodes and edges (*Myers, 1995*). As a result, the OLC method constructs a read graph that places reads as nodes and assigns a relationship between two nodes when these two reads overlap longer than the cutoff length. Paths through the graph are regarded as candidate contigs, and these paths can be translated into a series of genome sequences (*He et al., 2013*) as illustrated in Fig. 1. This process is further described in the next three steps.

1. The overlap of each pair of reads is identified using all-against-all pairwise read alignment. $K$-mers pre-calculation for all reads would improve performance considerably. It selects candidates that share $K$-mers and measures alignment by using $K$-mers as alignment seeds. The detection of overlaps is overly sensitive to limited overlap length and the size of the $K$-mer. The selection of these parameters can therefore significantly influence the assembler's efficiency. There would be too many candidates for small parameter values, while large values, in comparison, can lead to

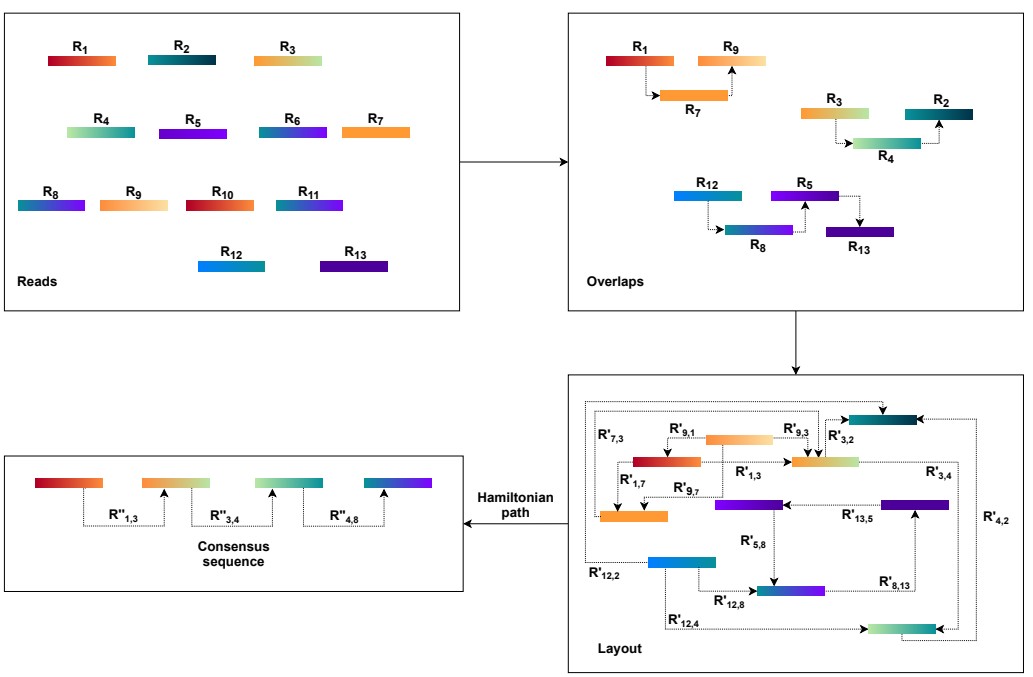

**Figure 1  The general workflow of OLC method.**

accurate contiguities that are shorter. Consequently, finding a good balance requires a considerable amount of time.

2. Based on the overlap information, the OLC constructs an overlap graph. Within this step, the OLC finds a special form of path, i.e., a simple path where every node is distinct. This path is a Hamiltonian path as the nodes will be visited exactly once. However, finding a Hamiltonian path is an NP-hard problem. This problem is solved in practice using a greedy strategy or heuristic algorithms.

3. Finally, the OLC performs multiple sequence alignment (MSA). MSA is intended to decide the exact layout and voting strategies. Alternatively, it may use statistical methods to define the best consensus sequence. However, no method efficiently resolves the optimal MSA problem. Therefore, the consensus stage uses pairwise alignments driven by the approximate read layout.

### Assemblers using OLC

Canu is a modern long read sequence assembler that strengthens and replaces the Celera Assembler (*Myers et al., 2000*; *Miller et al., 2008*). By integrating the MinHash Alignment Method (MHAP) with the PBcR (*Koren, 2012*) and Celera Assembler, the computational bottlenecks of overlapping noisy, single-molecule sequencing reads can be addressed (*Berlin et al., 2015*). Furthermore, Canu integrates these methods into one comprehensive assembler, which supports PacBio and Oxford Nanopore data, reduces the runtime and coverage needs, and improves the separation of repeats and haplotypes. It, therefore,

improves the mammalian genomes' efficiency by some degree and exceeds hybrid processes with as little as 20×single molecule coverage (*Koren et al., 2017*).

Canu (*Koren et al., 2017*) introduces several additional features to improve usability and efficiency with single-molecule sequence data, including computational resource discovery, adaptive $k$-mer weighting, automated error rate estimation, sparse graph construction, and graphical fragment assembly (*Li, 2016*) outputs. The Canu pipeline comprises three stages, each of which can be corrected, trimmed, and assembled in series or separately (*Koren et al., 2017*). Canu automatically senses available resources and configures itself to optimize the usage of resources when operating in parallel environments.

FASTA (*Lipman & Pearson, 1985*) or FASTQ (*Cock et al., 2010*) data is the primary data exchange,but for consistency, the input stores reads for each stage in an indexable database that no longer needs theoriginal input. Each of the three stages starts with the identification of overlaps between all input pairs. Although each stage has a different overlapping strategy, each one has an indexed store of these overlapsobtained by counting $k$-mers in the reads.

From the input reads, corrected reads are generated from the correction stage, unsupported bases and other anomalies are trimmed in the trimming stage, and assembly graphs and contigs in the assembly stage are finally built.

HINGE is another OLC assembler that aims to achieve an optimal resolution by differentiating between repeats that can be resolved and those that cannot (*Kamath et al., 2017*). To do this, hinges are applied to the reads for creating a graph where only unresolvable repeats are combined. Consequently, HINGE blends the error resilience of overlapping graphical methods, the elegant graph structure, and optimal repeat resolution of graphs by de Bruijn.

HINGE is a long read assembler that follows the paradigm of the overlap layout. Its key algorithmic advancement lies in how it uses the alignments achieved in the overlapping process to recognize resolvable repeats and build the structure of the graph repetitively (*Pevzner & Tang, 2001*; *Mulyukov & Pevzner, 2002*). To equip some of the reads with hinges, HINGE uses the alignment information collected during the overlap phase. The Contagion algorithm is utilized to disperse the information of how it bridges repeats to other reads (*Kamath et al., 2017*).

The Contagion algorithm allows HINGE to position precisely one in-and-out hinge on reads that have emerged from an unbridged repeat. Then, with a hinge-assisted greedy graph, HINGE can construct a sparse overlap graph. Given that our reads are hinged, as long as the match starts on the hinge, we also allow a read successor or predecessor to be within another read. Incidentally, the graph forms a bifurcation that corresponds to an unbridged repeat's beginning or end.

This hinge-aided approach helps us, within the OLC substructure, to achieve the attractive features of a de Bruijn graph.

## De Bruijn Graph (DBG) method

The De Bruijn Graph (*Flicek & Birney, 2009*; *Schatz, Delcher & Salzberg, 2010*; *Miller, Koren & Sutton, 2010*; *Compeau, Pevzner & Tesler, 2011*) method builds the whole-genome sequence of short reads. It utilizes $k$-mer graphs, suitable for large numbers of short reads.

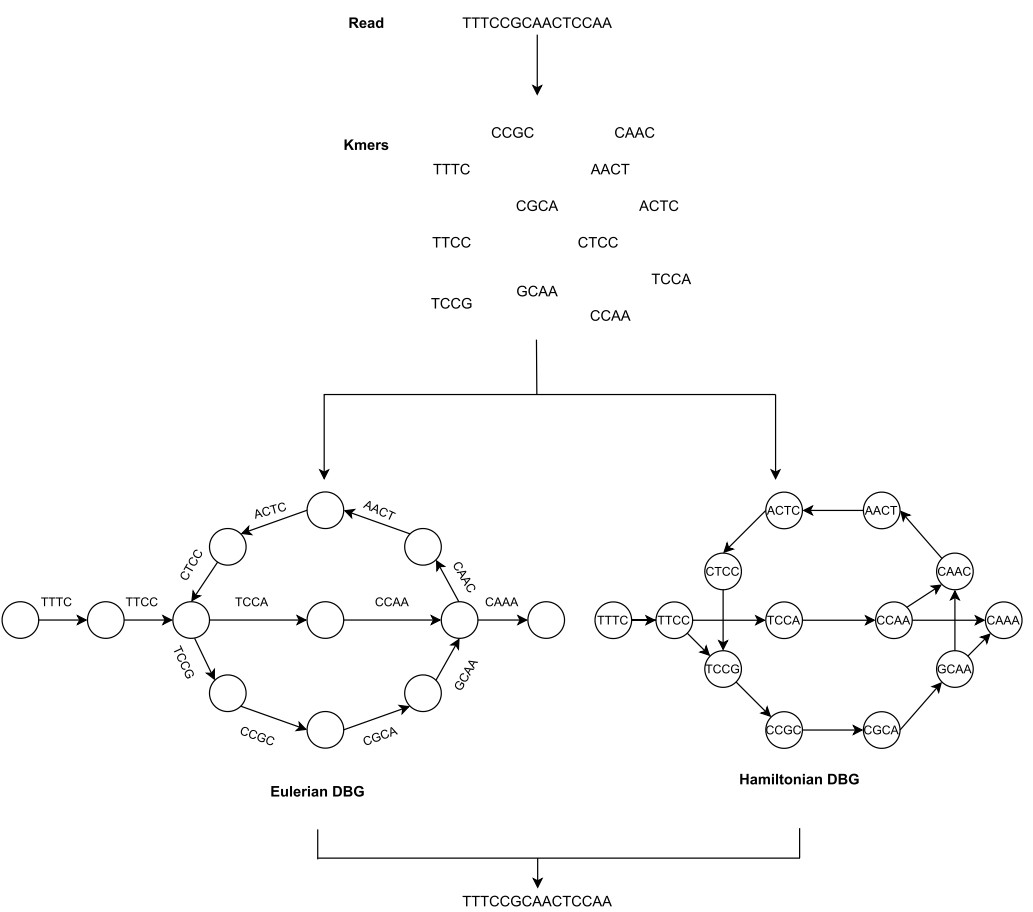

**Figure 2   The general workflow of DBG method.**

For the $k$-mer graph, it is no longer practical to consider all-against-all overlap. Each node represents a $k$-mer if an overlap of $k$-1 bases occurs and appears continuously in a read, and a directed arc will occur between the two nodes. There is also no need for individual reads and their overlap data to be saved (*He et al., 2013*). The overlap between adjacent $k$-mers is implicitly determined by cutting the reads into $k$-mers and recording their adjacent relationships. To summarize, the DBG approach generates a $k$-mer graph that considers $k$-mers to be nodes and assigns an edge to two nodes in the genome sequence where they are neighbors as illustrated in Fig. 2.

   Assembly methods based on de Bruijn graphs begin with the replacement of each read with the set of all overlaps of a shorter fixed length (*Liao et al., 2019*; *Chaisson, Wilson & Eichler, 2015*). Usually, $k$ denotes the length of the $k$-mers sequences.

   The value of $k$ is significant for constructing the de Bruijn graph (*Luo et al., 2015*). Some short redundant areas will be removed by a large value of $k$, thus reducing the number of nodes in the de Bruijn graph (*Benoit et al., 2014*), but this will induce disconnected subgraphs. A small value of $k$ will minimize those gap areas, thus increasing the connectivity

of the de Bruijn graph and adding additional nodes and increasing short recurring regions. The value of $k$ cannot, therefore, be too big or too small.

Adjacent to the reads, which cease at $k$-mers from repeat borders, are used to sculpt contigs. This requires very precise reads, which initially reduces the potential for reads to solve repeats longer than $k$-bases. It has the advantage that it does not require pairwise overlap storage and a graph structure that represents the genome's repeat structure. The following are the steps to be performed:

- Select a value for $k$
- Make $k$-mers
- Count the $k$-mers
- Make the DBG
- Categorize the de Bruijn graph based on the expressions of nodes and edges in two forms, i.e., Hamiltonian and Eulerian graph approach. The $k$-mers are the nodes in the Hamiltonian approach, while they are the edges in the Eulerian approach. The graph method in the Hamiltonian approach resembles the OLC method. The sequences are constructed in this approach to find Hamiltonian paths that pass through all nodes and are only visited once. This is an NP-complete problem when the number of nodes is not negligible. Consequently, this makes assembly problems a simpler issue in the theory of algorithms, which is the most crucial advantage of DBG.
- Revise contigs from the simplified graph.

### Assemblers using DBG

SOAPdenovo (*Li et al., 2010*) has been used to compile several genomes successfully but its continuity, accuracy, and coverage, in repeat regions in particular, need to be enhanced. To overcome this difficulty, SOAPdenovo2 (*Luo et al., 2012*) has been built to solve more repeated contiguous areas, increase coverage and length in scaffolding, boost gap closure, and optimize for the large genome. The current architecture of the SOAPdenovo2 algorithm reduces the memory consumption for graph construction (*Ye et al., 2012*).

SOAPdenovo2 substantially improves: (1) the algorithm for error correction, (2) memory usage in DBG constructions, (3) assembly length and scaffolding coverage, (4) the closing of gaps, and (5) resolution of longer repeat areas in contig assemblies.

SOAPdenovo2 consists of six modules, like SOAPdenovo. (1) Genome DNA is randomly fragmented using paired-end technology and sequenced. Short read with sizes between 150 and 500bp are amplified and sequenced directly, whereas long-range (2–10 kb) paired-end libraries are constructed by circularizing DNA, fragmentation, and then cleansing fragments for cluster creation with ranges of 400–600 bp. (2) To represent the overlap between the reads, the raw or corrected reads are then loaded into computer memory, and de Bruijn graph data structure is used (*Li et al., 2010*). (3) By eliminating erroneous ties and resolving small repeats by read paths, the graph is simplified as follows: (a) Cut-off of short tips, (b) Deletion of links with low coverage, (c) Resolution of tiny repeats with a read path, and (d) fusion of bubbles formed by repeats or heterozygotes of diploid chromosomes. (4) The links at repeat boundaries are broken on the simplified graph, and unambiguous sequence fragments are output as contigs. (5) By using paired-end information, reads are

realigned to contigs, and scaffolds are generated from unique contigs. (6) Finally, using the paired-end extracted reads, intra-scaffold gaps are filled.

SPAdes (*Bankevich et al., 2012*) is a universal A-Bruijn assembler (*Pevzner, Tang & Tesler, 2004*) for single and multi-cell assembly. It performs an operation that does not directly affect the sequences; rather, it performs graph topology, coverage, and length of the sequence. It uses $k$-mers for the sole purpose of constructing the de Bruijn graph. It performs graph-theoretical operations on the subsequent stages exclusively in graphs that need not be labeled by $k$-mers (*Bankevich et al., 2012*). The consensus sequence of DNA is restored at the last stage.

During reconstruction, a string from the set of its $k$-mer is often abstracted in fragment assembly. The de Bruijn approach to assembly leads to this abstraction, which underlies several algorithms for assemblies. However, a more progressive abstraction of NGS data takes into account the problem of reforming a string from a set of pairs of $k$-mers ($k$-bimers) at a distance approximately d in a string (*Bankevich et al., 2012*). The study of the latter abstraction has chiefly been associated with heuristic post-processing, even though there are simple algorithms available for the former abstraction in the de Bruijn graph (*Pevzner, Tang & Waterman, 2001*; *Zerbino & Birney, 2008*; *Butler et al., 2008*).

By adding $k$-bimer modification, which concedes the exact distances for the vast majority of the adjusted K-bimers (*Bankevich et al., 2012*), SPAdes solves the impracticality due to differences in the biread1 (and thus $k$-bimer) distances and adding PDBG-inspired paired assembly graph (*Medvedev et al., 2011*). SPAdes may use read-pairs; in particular, E+V-SC (*Chitsaz et al., 2011*) has been using the reads but has skipped the read-pairs pairing to prevent misassemblies because of a high level of chimeric read-pairs.

The four stages of SPAdes that resolve problems that are especially problematic in SCS are sequencing errors; non-uniform coverage; disparities in insert size; and chimeric reads and bireads.

- The accurate distance estimation performed in this stage ($k$-bimer adaptation) between $k$-mers is based upon the joint distance histogram and assembly graph analysis.
- Inspired by the PDBG (*Medvedev et al., 2011*) method, the paired assembly graph is constructed in this stage (contig construction).
- The DBG is constructed.
- By backtracking graph simplifications, the last stage (contig construction) is completed. SPAdes generates DNA sequences of contigs and maps reads to contigs.

## String graph-based method

The string graph is a simplified version of a classic overlap graph with sequenced reads and a suffix to prefix overlaps with the non-transitive edges (*Liao et al., 2019*). The string graph is an essential data representation used by OLC assemblers. Indeed, the vertices in a string graph are the input reads, and the arcs correspond to the overlapping reads, which are contigs in the string graph. For long-read assembly, an overlap-based approach is a forthright approach because it assembles the long reads without being translated to $k$-mers.

The formulation of the string graph assembly is similar to a de Bruijn graph in principle. However, it has the advantage of not decomposing sequences into $k$-mers, but taking the

complete length of a read sequence (*Liao et al., 2019*). From the overlap graph, the string graph can be extracted by first removing duplicate reads and contained reads, and then discarding transitive edges from the graph.

For long sequences and single-molecule sequencing reads with a high error rate, the overlap-based approaches are more acceptable than the de Bruijn graph-based methods.

### Assemblers using string graph

SGA is an assembler based on FM-index (*Ferragina & Manzini, 2005*) derived from the compressed Burrows–Wheeler transform (*Burrows & Wheeler, 1994*), memory-efficient data structures, and assembly algorithms (*Simpson & Durbin, 2012*). In comparison to most *de novo* assemblers, which depend on de Bruijn graphs, the SGA model uses the overlap string graph, which can easily be paralleled.

As *de novo* assembly usually demands queries over the entire sequence, extensive datasets tend to be a practical problem for assembly software developers and users. The redundancy contained in a sequence is exploited using compressed data structures to reduce the memory needed to perform *de novo* assemblies.

The SGA algorithm is based on an FM-index query developed from a set of sequence reads. The SGA pipeline starts with various low-quality or ambiguous base calls by preprocessing the sequence reads to filter or trim reads (*Simpson & Durbin, 2012*). From the filtered set of reads, the FM-index is constructed and base-calling errors are detected and corrected using $k$-mer frequencies. Corrected reads are re-indexed and duplicate sequences are discarded, filtering out the remaining low-quality sequences and generating a string graph. Contigs, if paired-end or mate-pair data is available, are assembled from the string graph and built into scaffolds.

SGA provides the first functional assembler, to the best of our knowledge, of a mammalian-sized genome on a low-end computing cluster, given its low memory requirements and parallelization without requiring inter-process communication.

FALCON, a long-read assembler with perceptive analysis of diploid genomes, is designed to assemble haplotype contigs that represent the diploid genome with correctly phased homologous chromosomes (*Chin et al., 2016*). It also preserves ambiguity in the assembly graph and outputs the longest path through the graph along with alternate paths (*Liao et al., 2019*; *Koren & Phillippy, 2015*).

The FALCON assembler follows the hierarchical genome assembly process(HGAP) (*Chin et al., 2013*) design but uses components that are more computationally optimized. To create a string graph containing sets of 'haplotype-fused' contigs and bubbles representing divergent regions between homologous sequences, it begins by using reads. Next, using phase data from heterozygous positions that it identifies, FALCON-Unzip identifies read haplotypes. Phased reads are then used with phased single-nucleotide polymorphisms and structural variants to assemble haplotype contigs and primary contigs that form the final diploid assembly.

As compared to alternative short or long-read approaches, the FALCON-based assemblies are significantly more contiguous and complete. The phased diploid assembly capacitated the analysis of the structure of the haplotype and heterozygosities between

homologous chromosomes, including the identification within coding sequences of widespread heterozygous structural variation.

Hifiasm, a modern *de novo* assembler that faithfully represents haplotype information in a phased assembly graph by using long high-fidelity sequence reads (*Cheng et al., 2021*). Hifiasm aims to maintain the contiguity of all haplotypes, unlike other graph-based assemblers that only seek to maintain the contiguity of one haplotype. This function allows for the development of a graph trio binning algorithm that is superior to regular trio binning.

Hifiasm corrects sequence errors while maintaining heterozygous alleles using haplotype-aware error correction and then builds phased assembly graphs using locally corrected reads for phasing information. In the phased assembly graph, only reads from the same haplotype are linked. hifiasm produces a fully phased assembly for each haplotype from the graph using complementary data that provides global phasing information. Only HiFi reads can be used by Hifiasm to produce an unphased primary assembly. This unphased primary assembly constitutes the phase blocks (regions), which can be solved with HiFi reads but cannot maintain phase information between two-phase blocks.

Hifiasm's first few steps are relatively similar to the workflow of early long-read assemblers. Hifiasm performs an overlap alignment of all-vs-all and then corrects sequencing errors. hifiasm inspects the alignment of reads overlapping with the target read when given a target read to correct. An informative position on the target read is said to be provided if at the alignment two types of A/C/G/T bases are in place and if at least three reads support each type. If there are informative positions in the overlap and the read is not identical to the target read in all of these positions, the read is inconsistent with the target. Only clear reads are used by Hifiasm to correct the target read.

By default, Hifiasm performs three rounds of error correction. It then performs overlap alignment once more and constructs a string graph with a vertex representing an oriented read and an edge representing a consistent overlap. A pair of heterozygous alleles in the string graph will be represented by a bubble after transitive reduction. There is no data loss. If no additional data is available, hifiasm chooses one side of each bubble at random and produces a primary assembly, similar to Falcon-Unzip (*Chin et al., 2016*) and HiCanu (*Nurk et al., 2020*).

## Hybrid method
### Assemblers using hybrid methods
The chemistry of Illumina has improved dramatically since the release of A5-MiSeq instruments are now able to generate reads above 400nt in length, which is four times longer than what was previously possible on the HiSeq 2000. The initial A5 was unable to process reads longer than 150 nt. The longer reads make it possible to assemble genomes from fewer data in general, but significant revisions to the data processing algorithms in A5 were needed to do so (*Coil, Jospin & Darling, 2015*).

A revised A5-MiSeq pipeline, which substitutes new software modules for many components of the original A5 pipeline, produces dramatically improved assemblies. The A5-MiSeq pipeline comprises five steps (*Coil, Jospin & Darling, 2015*): read cleaning,

contig assembly, crude scaffolding, misassembly correction, and final scaffolding. The details are as follows: (i) Trimmomatic (*Lohse et al., 2012*) eliminates sequence adapters and low-quality regions. The SGA *k*-mer-based error correction algorithm (*Simpson & Durbin, 2012*) then corrects errors in the reads. (ii) The IDBA-UD algorithm (*Peng et al., 2012*) is used to assemble paired and unpaired reads.(iii) Contigs are scaffolded using permissive parameters by any large insert library available. (iv) Misassemblies are detected by read pairs not mapped within the intended distance. Contigs and scaffolds that include misassemblies are broken. (v) A final round of scaffolding repairs any previously broken contiguity with strict parameters.

A5-MiSeq revises steps I and II considerably compared to A5. All the modifications lead to dramatically enhanced assemblies that recover a more complete set of reference genes than previous methods.

Flye (*Kolmogorov et al., 2019*) is a long-read assembly algorithm that constructs an accurate repeat graph from the arbitrary paths it produces called disjointigs. Flye constructs the repeat graph from error-riddled disjointigs.

Initially, Flye produces disjointigs representing several disjointed genome segment concatenations. All error-prone disjointigs are joined into a single string. The resulting string is used to construct an accurate assembly graph. Then reads are used to untangle the graph and resolve bridged repeats. Later, the repeat graph uses minor variations between repeat copies to address unbridged repeats (which are not bridged by any reads) and then outputs precise contigs formed by paths in this graph (*Kolmogorov et al., 2019*).

Assembly graphs are a special case of breakpoint graphs (*Chin et al., 2016*) and hence are well suited to examine structural variations (*Koren & Phillippy, 2015*; *Coil, Jospin & Darling, 2015*) and segmental duplications (*Phillippy, Schatz & Pop, 2008*; *Nagarajan & Pop, 2009*). Flye assembly graphs provide a valuable framework for segmental duplications to be reconstructed and additional genome completion experiments to be planned.

## EXPERIMENTS

The same set of datasets was given to all assemblers in our evaluation. The experiments were all carried out on the same server (two Intel Xeon Processor E5-2695 v4) with a limitation of 128GB memory. We begin with a description of the dataset. Before presenting the results of various evaluations, we outline the assemblies.

### Dataset

In six projects covering three bacteria, a mammal, a plant, and a fungus, we have chosen whole-genome shotgun data of *Arabidopsis thaliana*, *Bacillus cereus*, *Caenorhabditis elegans*, *Escherichia coli*, *Saccharomyces cerevisiae*, and *Staphylococcus aureus*. We have also included a human genome to see how versatile assemblers are with big genomes. All the species have previously been sequenced and completed using one of the above assemblies to a very high level. With the previously sequenced genome, the correctness of each assembler has been stringently evaluated and compared.

A wide range of genome sizes is also expressed by the seven genomes, from bacteria to human genome. This smaller sample was selected because some of the assemblers would

**Table 1  Dataset information.**

| Dataset | Read type | Technology | Accession number | Refseq | # of bases | Coverage |
|---------|-----------|------------|------------------|--------|------------|----------|
| *Arabidopsis thaliana* | Short | Illumina | ERR3485043 | GCF_000001735.4 | 304.3M | 2.3 |
|  | Long | PacBio | ERR3415827 |  | 1.9G | 8.7 |
| *Bacillus cereus* | Short | Illumina | ERR3338758 | GCF_000007825.1 | 443.6M | 3.0 |
|  | Long | PacBio | SRR9641613 |  | 1.2G | 25.6 |
| *Caenorhabditis elegans* | Short | Illumina | SRR12431876 | GCF_000002985.6 | 373.8M | N/A |
|  | Long | PacBio | ERR3489110 |  | 1.5G | N/A |
| *Escherichia coli* | Short | Illumina | SRR12573761 | GCF_000005845.2 | 326.7M | 62.9 |
|  | Long | PacBio | SRR10538960 |  | 3.3G | 488.9 |
| Human | Short | Illumina | SRR005721 | GCF_000001405.39 | 860.9M | N/A |
|  | Long | PacBio | SRR13684281 |  | 6.8G | 2.3 |
| *Saccharomyces cerevisiae* | Short | Illumina | SRR12596359 | GCF_000146045.2 | 3.0G | 225.0 |
|  | Long | PacBio | ERR4467305 |  | 5.3G | 288.7 |
| *Staphylococcus aureus* | Short | Illumina | SRR12560295 | GCF_000013425.1 | 480.7M | 167.8 |
|  | Long | PacBio | SRR10807892 |  | 2.5G | 715.5 |

**Notes.**

All the reads are taken from NCBI.

take several weeks to assemble the whole genome, and others would ultimately fail. Table 1 summarizes the details of reads used for experiment.

## Assemblies

To allow comparisons between assemblies of different assemblers, we run the assembler under the default parameters except for assembling the human genome. Default parameters were not adequate to let the assemblers run on the human genome; therefore, we fine-tuned the parameters of each assembler. To decide the best assembly for each assembler, without consideration of assembly errors, we used the contig (N50, NG50, and genome fraction) sizes as the primary metric. This method is similar to what typically happens between groups that assemble a new genome: assembly with the largest contigs and scaffolds is generally favored.

The data cleaning procedure is one of the most critical phases in any assembly and often takes considerably longer than the assembly. Genome data is never flawless, and the various types of errors will cause various assembly problems. We did not want to differentiate the efficacy of error correction and the assembly algorithms themselves; some of the assemblers that we ran have their own built-in error correction routines. Therefore, if the assembler comprises one, the first step we ran for each of the datasets was an error correction procedure. When using their error correction routines, most of the assemblers were the most effective. If a dataset does not run on an assembler, the results will not be included for the assembler.

We used a few metrics to present snapshots of each assembly: number of contigs, largest contig, size of N50, NA50, NG50, and GC content of contigs are some of the metrics. The N50 value is the contig length such that half (50%) of the assembly bases are generated by using longer or equal length contigs (*Gurevich et al., 2013*). There is typically no value that

produces precisely 50%, so the technical meaning is the maximum length x to account for at least 50% of the overall assembly length by using contigs of length x or higher. NG50 is the length of the contigs, which generates 50% of the bases of the reference genome with longer or equal length of the contig, whereas NA50 is N50, where aligned block lengths rather than contigs lengths are counted. In other words, the contig is split into smaller parts when there is a misassembly with the reference genome. The latter two metrics can only be determined by providing a reference genome. Determined by the total length of the assembly, the GC is the total number of g and c nucleotides in the assembly.

Assemblers may also be determined based on how well-known sequences are retrieved from the respective assembly. By aligning assemblies with a completed reference genome, we evaluated the correctness of the assemblies.

There are many common assembly issues: several small contigs, missing sequences, needless duplication of contiguous contigs, and misassembly errors. Some of these errors are unique to individual assemblies, and some are endemic. An estimate of repeat copy numbers is one of the more complicated aspects of genome assembly. A duplicated repeat is one that occurs in more copies in the assembly than required. It is a preventable issue that many of the assemblers address better than others. In the *Staphylococcus aureus* dataset, for instance, Canu and Hinge, Canus N50 is threefolds of that of HINGE. However, depending on how much of the N50 is correctly compatible (NGA50) with the reference genome, HINGE is better than Canus. This is owing to the duplication in the Hinge assembly of contiguous contigs.

The overall length of the majority of the assemblies presented was marginally higher than the size of the genome, mostly owing to the polymorphic degree. Usually, the assembly-based N50s are smaller than the NG50s. Such differences are diminutive but not always negligible: for instance, a 58 kb NG50 longer than N50 is available on the *Escherichia coli* assembly.

Indels (insertions and deletions) also differ based on assembler. Based on the objective of the assembler, the number of indels might differ. However, as we use a reference genome and read set, the relative number of indels should be a reliable measure of errors between assemblers.

If an assembler's objective is to increase contigs lengths, it will be susceptible to producing more indels. If its objective is to eliminate errors, then the length of the contigs it creates will be small. For instance, on the *Arabidopsis thaliana* data, FALCON strives to maximize its contig length at the expense of indels. FALCON generates more indels than the rest of the assemblers, while SGA minimizes indels at the expense of contig length. Even though FALCON produces more indels, its mismatch level is approximately as good as assemblers that generate small indels.

Structural errors are a harmful type of error; as described in ''Introduction' misassembly plays a significant role in causing such errors. We used QUAST to overcome the problem. Misassembly errors are divided into extensive or local as per QUAST's definition. The following condition has been described as an extensive misassembled contig: (i) the left flanking sequence aligns on the reference over 1 kb away from the right flanking sequence; (ii) overlaps of flanking sequences are more than 1 kb, and (iii) the flank sequence aligns

**Table 2** Assembly statistics of assemblers on *Arabidopsis thaliana* dataset.

| Read type | Assemblers | # Contigs | N50 | L50 | GC(%) |
|-----------|-----------|-----------|-----|-----|-------|
| **Long** | A5-MiSeq | 11,739 | 637 | 7876 | 36.65 |
| | Canu | 162 | 10,862 | 48 | 43.09 |
| | Falcon | 735 | 10,115 | 225 | 40.95 |
| | Flye | 1,826 | 77,106 | 860 | 37.24 |
| | Hinge | 522 | 29,990 | 145 | 41.02 |
| | SGA | 98 | 10,612 | 7 | 38.14 |
| **Short** | SPAdes | 26,750 | 722 | 9,629 | 36.8 |
| | SOAPdenovo2 | 31 | 89,813 | 1 | 35.94 |

**Notes.**
   The minimum number of contigs generating 50% of the assembly base is represented by L50.

with different strands or different chromosomes, while the local misassembled contigs fulfill the following: (i) breakpoint is covered by multiple distinct alignments; and (ii) the left and right flanking sequences are on the same strand on the same chromosome of the reference genome, and the gaps between them are less than 1 kb. Misassembly errors can be mistaken as true genetic modifications because actual biological differences are similarly manifested. Therefore, it is important to recognize these errors. Three forms of misassemblies have been cataloged: inversion, relocation, and translocation. In parallel with the NG50 value, we used the NGA50 so that the misassemblies are considered for assessing the assemblies. Notice that some NGA50 values are undefined: in *Caenorhabditis elegans*, for example, misassemblies occur when all aligned blocks have a total length of less than 50% of the total assembly length.

## RESULTS

The tables below present very significant performance differences between assemblers, as well as variations in an individual assembler's performance when applied to different genomes. The following are some cases of assembler performance.

### *Arabidopsis thaliana*

It can be seen from Table 2 that the short read assemblers produced more contigs and had a higher N50 value than the long read assemblers for *Arabidopsis thaliana*. SOAPdenovo2 produces much larger contigs than any other assembler, with an N50 size of 89.8 kb, while SPAdes produces the largest contig of 26.7 kb. After comparing SOAPdenovo2 to the reference genome (Table 3), we discovered that it has many assembly errors: its total number of uncalled bases (N's) was the highest in the assembly. Analyzing the assembly at these points yielded no result for the value of NG50. However, at the cost of indels and mismatches, SPAdes generated the longest contigs.

The N50 value for Flye was 77 kb, and the breaking of the contigs lowered the NG50 value less drastically to 68 kb with far fewer uncalled bases, making it the best of the assemblers on this genome . SPAdes with a lower N50 value than SOAPdenovo2, and their NG50 was the same (i.e., undefined), but SPAdes appeared to be preferable to SOAPdenovo2 with around half of the assembly errors (Fig. 3 and Table 4).

**Table 3  Assembly statistics of datasets with a reference genome.** NGA50 is NG50 in which aligned block lengths are counted rather than contig lengths. LA75 follows the same analog as NGA50 concerning L70. GF is genome fraction. DR is duplication ratio. LA is largest alignment and TAL is total aligned length.

| Data | Assemblers | Gf (%) | DR | LA | TAL | NG50 | NA50 | NGA50 | LA75 | Time |
|------|-----------|--------|-----|-----|------|------|------|-------|------|------|
| *Arabidopsis thaliana* | A5-MiSeq | 6.496 | 1.027 | 55632 | 7821611 | - | 618 | - | 8163 | 70 |
| | Canu | 0.5 | 3.624 | 103154 | 1957236 | - | 8557 | - | 145 | 698 |
| | Falcon | 2.785 | 1.753 | 52449 | 5753720 | - | 7414 | - | 605 | 662 |
| | Flye | 71.501 | 1.029 | 323229 | 87783639 | 68321 | 37683 | 31223 | 2014 | 147 |
| | Hinge | 0.52 | 1.748 | 72349 | 1082572 | - | - | - | - | 295 |
| | SGA | 0.151 | 1.005 | 17368 | 181086 | - | 10612 | - | 22 | 10 |
| | SOAPdenovo2 | 0.119 | 1.093 | 84661 | 143517 | - | 84661 | - | 4 | 0.17 |
| | SPAdes | 15.989 | 1.042 | 31938 | 19105410 | - | 606 | - | 21082 | 17 |
| *Bacillus cereus* | A5-MiSeq | 5.761 | 1.014 | 31356 | 316547 | 94223 | - | - | - | 70 |
| | Canu | 18.489 | 1.096 | 15861 | 1099326 | 1271827 | - | - | - | 75 |
| | Falcon | 17.567 | 1.486 | 30007 | 1415685 | 34235 | - | - | - | 260 |
| | Flye | 17.877 | 1.022 | 63366 | 991230 | 3779838 | - | - | - | 17 |
| | Hinge | 21.753 | 1.544 | 10569 | 1823312 | 2010834 | - | - | - | 41 |
| | SGA | 5.924 | 1.031 | 31081 | 329564 | 18389 | - | - | - | 12 |
| | Spades | 5.602 | 1.71 | 13119 | 304717 | 24307 | - | - | - | 11 |
| | Soapdenovo2 | 5.632 | 1.009 | 31143 | 307668 | 94674 | - | - | - | 0.73 |
| *Escherichia coli* | A5-MiSeq | 91.246 | 1.004 | 194019 | 4238262 | 207510 | 46684 | 55105 | 69 | 240 |
| | Canu | 89.572 | 1.013 | 146928 | 4211826 | 4056254 | 24795 | 51958 | - | 22 |
| | Falcon | 48.749 | 2.294 | 20252 | 5152719 | 9582 | 2499 | 4511 | - | 1040 |
| | Flye | 89.557 | 1.011 | 146930 | 4201449 | 1349247 | 28464 | 51984 | - | 52 |
| | Hinge | 89.623 | 2.012 | 147472 | 8360148 | 3358761 | 28410 | 72928 | - | 565 |
| | SGA | 91.188 | 1.004 | 193965 | 4237989 | 196903 | 52805 | 55100 | 62 | 30 |
| | Spades | 90.968 | 1.009 | 194101 | 4227658 | 216146 | 54887 | 55987 | 63 | 7 |
| | Soapdenovo2 | 91.082 | 1.002 | 193904 | 4224949 | 196831 | 41435 | 52593 | 77 | 26 |
| Human genome | A5-MiSeq | 2.437 | 1.015 | 18785 | 75924817 | - | 583 | - | 86057 | 1075 |
| | Canu | 1.383 | 1.421 | 108455 | 59774470 | - | 10241 | - | 5608 | 4320 |
| | Falcon | 0.874 | 1.012 | 64590 | 27427277 | - | 12924 | - | 1409 | 631 |
| | Flye | 0.866 | 1.03 | 93090 | 27601994 | - | 26154 | - | 762 | 67 |
| | HiFiasm | 10.488 | 1.014 | 74358 | 330374434 | - | 21579 | - | 10562 | 17 |
| | Hinge | 0.207 | 2.197 | 60176 | 13733461 | - | 1386 | - | - | 35 |
| | SGA | 0.013 | 1.039 | 7137 | 401793 | - | 596 | - | - | 506 |
| | Spades | 0.005 | 1.064 | 16600 | 166272 | - | 827 | - | 160 | 35 |
| *Saccharomyces cerevisiae* | A5-MiSeq | 93.834 | 1.004 | 238989 | 11422884 | 87302 | 77574 | 72006 | 98 | 294 |
| | Canu | 95.269 | 1.144 | 546941 | 13088915 | 789964 | 160884 | 199769 | 56 | 229 |
| | Falcon | 3.176 | 10.781 | 17395 | 4111569 | - | 3172 | - | - | 2036 |
| | Flye | 94.704 | 1.018 | 546784 | 11700248 | 904738 | 230022 | 218721 | 38 | 65 |
| | Hinge | 93.224 | 1.943 | 532988 | 22012911 | 1015480 | 195155 | 337524 | 83 | 79 |
| | Spades | 93.195 | 1.028 | 538406 | 11406775 | 234358 | 149088 | 147545 | 48 | 39 |
| | Soapdenovo2 | 93.518 | 1.003 | 328324 | 11377122 | 109730 | 102238 | 93828 | 79 | 5 |

**Table 3** (*continued*)

| Data | Assemblers | Gf (%) | DR | LA | TAL | NG50 | NA50 | NGA50 | LA75 | Time |
|------|-----------|--------|-----|-----|-----|------|------|-------|------|------|
| | A5-MiSeq | 88.836 | 1.006 | 171163 | 2518098 | 177125 | 72014 | 72014 | 30 | 71 |
| | Canu | 92.505 | 1.032 | 329791 | 2692695 | 2907970 | 92196 | 92541 | 21 | 61 |
| | Falcon | 89.225 | 6.2 | 24089 | 15571683 | 15602 | 4915 | 11704 | 2471 | 817 |
| *Staphylococcus aureus* | Flye | 92.502 | 1.005 | 329938 | 2618043 | 2896520 | 92541 | 92541 | 19 | 58 |
| | Hinge | 92.561 | 2.033 | 323857 | 5303669 | 1311923 | 91667 | 151987 | 44 | 157 |
| | SGA | 88.509 | 1.008 | 171354 | 2509007 | 109236 | 54884 | 54884 | 35 | 11 |
| | Spades | 86.544 | 1.057 | 259183 | 2442671 | 180321 | 91655 | 103869 | 25 | 35 |
| | Soapdenovo2 | 88.562 | 1.003 | 171154 | 2504468 | 174162 | 72014 | 72014 | 30 | 0.36 |

**Notes.**
NGA50 is NG50 in which aligned block lengths are counted rather than contig lengths. LA75 follows the same analog as NGA50 concerning L70. GF is genome fraction. DR is duplication ratio. LA is largest alignment and TAL is total aligned length. All assemblers except SOAPdenovo2 and SPAdes represents long read assembler.

### *Bacillus cereus*

The largest contigs was generated by Flye (with an N50 size of 3.7 Mb), followed by Hinge (821 kb) and Canu (858 kb). Falcon was one of the most error-prone assemblers, with 15 misassemblies and 647 indels for *Bacillus cereus*, and it only had an NG50 value of 34.2 kb after the error correction. Hinge has had almost as many errors and rose even more to NG50 of 2 Mb. With NG50 the same as their N50 value, Canu, Flye, and SPAdes had fewer errors.

It can be seen from Table 5 that SOAPdenovo2 has the fewest contigs for *Bacillus cereus*, as well as a low N50 value. Even though it produced the fewest contigs, its error rate was among the best amongst those assemblers, resulting in the fourth highest NG50 value of 94.6 kb. While SPAdes has fewer errors, it has the highest duplication ratio, resulting in a low NG50 value.

Flye's contiguity remained high, and its NG50 of 3.7 Mb was the highest after correction, followed by Hinge with an NG50 of 2 Mb. Although Flye has the highest NG50 value when compared to the reference genome, it is the third-best assembler for *Bacillus cereus* genome, behind Hinge and Canu (Table 3); we noticed that it contained the largest misassembled contigs length.

### *Caenorhabditis elegans*

With the exception of Flye producing the highest N50 value, Flye, SOAPdenovo2, and SPAdes has the smallest number of contigs for *Caenorhabditis elegans*. The performance of SOAPdenovo2 and SPAdes on this genome was poor. Their largest contigs and N50 values were less than 1kb, making it an unpreferable assembler for *Caenorhabditis elegans*, compared to SOAPdenovo2 and SPAdes.

There were other assemblers, however, such as Canu and Flye, with higher N50 values. We ran this data set without a reference, as mentioned earlier in 'Experiments'. To determine the performance of the assemblies, we chose the N50 value as a primary metric. Canu produced an N50 of 5.8 Mb with these metrics, being next to 5.9 Mb of N50 value of Flyes, as summarized in Table 6.

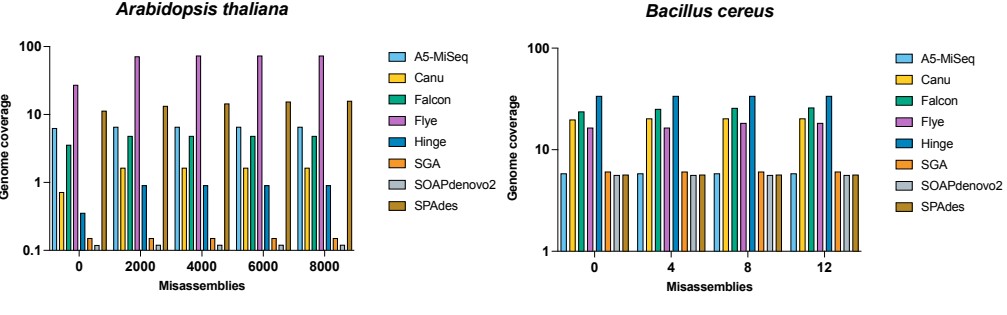

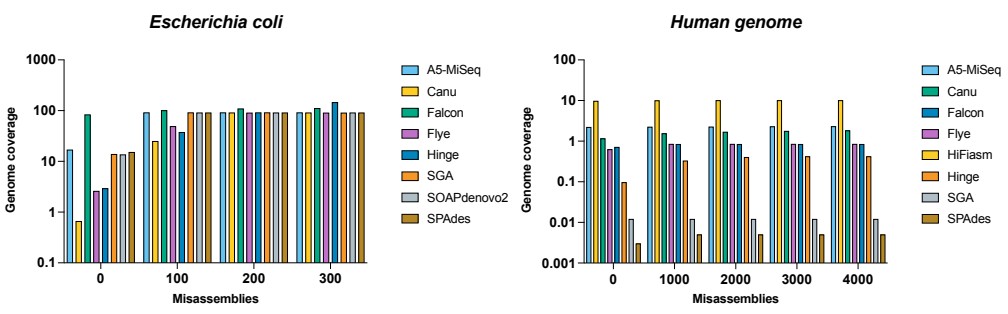

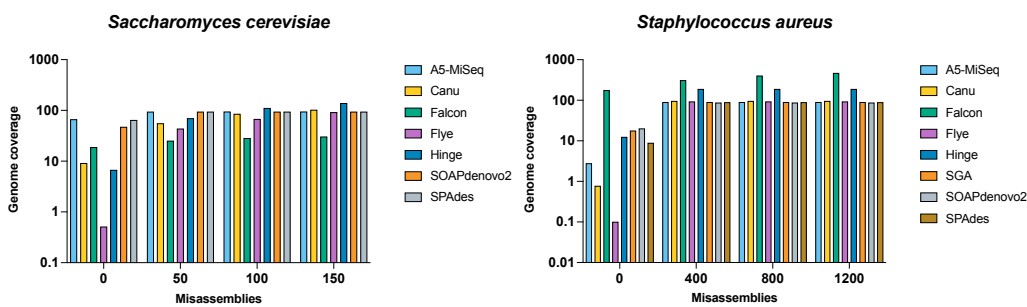

**Figure 3** **Comparison of misassemblies of datasets with each assemblers.** The *Y* axis is the total number of aligned bases divided by the reference length, in the contigs having the total number of misassemblies at most *X*. All assemblers except SOAPdenovo2 and SPAdes represents long read assembler.

### Escherichia coli

Falcon, SPAdes, and A5-MiSeq generated the highest number of contigs for *Escherichia coli*, while Canu, Flye, and Hinge generated the longest contig in the assembly. In comparison to assemblies with the highest number of contigs, assemblies with the largest contigs have the highest N50 values. Falcon came in last with an N50 of 6.1 kb, followed by SPAdes with an N50 of 196 kb, and then A5-MiSeq with an N50 of 197 kb, as summarized in Table 7. Basically, as contig lengths grow longer, the assembler becomes more susceptible to errors

**Table 4  Unaligned and mismatched statistics of each datasets.**

| Data | Assemblers | Unaligned | | Mismatches | |
|------|-----------|-----------|---|-----------|---|
| | | # fully unaligned contigs | Fully unaligned length | # mismatches | # indels |
| *Arabidopsis thaliana* | A5-MiSeq | 188 | 120181 | 33388 | 4523 |
| | Canu | 0 | 0 | 49425 | 8398 |
| | Falcon | 1 | 4663 | 45626 | 69358 |
| | Flye | 1 | 4544 | 1350240 | 1438162 |
| | Hinge | 95 | 1145948 | 9363 | 20687 |
| | SGA | 0 | 0 | 8 | 2 |
| | SOAPdenovo2 | 0 | 0 | 163 | 12 |
| | SPAdes | 232 | 162162 | 157325 | 12794 |
| *Bacillus cereus* | A5-MiSeq | 93 | 751664 | 11700 | 117 |
| | Canu | 9 | 146028 | 37826 | 282 |
| | Falcon | 71 | 820032 | 49672 | 674 |
| | Flye | 4 | 93567 | 35165 | 256 |
| | Hinge | 5 | 161003 | 51871 | 1693 |
| | SGA | 519 | 1724075 | 11977 | 108 |
| | SOAPdenovo2 | 359 | 1477993 | 11590 | 99 |
| | SPAdes | 66 | 636544 | 11508 | 98 |
| *Escherichia coli* | A5-MiSeq | 96 | 236146 | 52596 | 812 |
| | Canu | 82 | 451545 | 92909 | 1309 |
| | Falcon | 375 | 1870266 | 110250 | 3622 |
| | Flye | 9 | 165451 | 92818 | 1327 |
| | Hinge | 18 | 350499 | 191453 | 28035 |
| | SGA | 51 | 56900 | 52556 | 856 |
| | SOAPdenovo2 | 22 | 117911 | 52189 | 794 |
| | SPAdes | 350 | 432044 | 52367 | 782 |
| *Human genome* | A5-MiSeq | 1519 | 1327725 | 176747 | 25981 |
| | Canu | 398 | 4419980 | 729243 | 65863 |
| | Falcon | 1 | 136597 | 310378 | 98715 |
| | Flye | 42 | 383749 | 342934 | 77101 |
| | HiFiasm | 20 | 587213 | 541861 | 712152 |
| | Hinge | 147 | 2007596 | 281097 | 29752 |
| | SGA | 109 | 175934 | 1620 | 83 |
| | SPAdes | 15 | 20781 | 2253 | 107 |
| *Saccharomyces cere-visiae* | A5-MiSeq | 39 | 38622 | 63755 | 5780 |
| | Canu | 1 | 1782 | 125881 | 16218 |
| | Falcon | 3 | 48343 | 73016 | 15369 |
| | Flye | 0 | 0 | 75321 | 11172 |
| | Hinge | 22 | 150873 | 193317 | 491909 |
| | SOAPdenovo2 | 9 | 10420 | 64077 | 6005 |
| | SPAdes | 58 | 48546 | 65866 | 5594 |

**Table 4** (*continued*)

| Data | Assemblers | Unaligned | | Mismatches | |
|------|-----------|-----------|-----------|-------------|-----------|
| | | # fully unaligned contigs | Fully unaligned length | # mismatches | # indels |
| | A5-MiSeq | 2 | 3052 | 34933 | 1233 |
| | Canu | 0 | 0 | 5337 | 379 |
| | Falcon | 23 | 188293 | 39852 | 21850 |
| *Staphylococcus au-reus* | Flye | 0 | 0 | 5024 | 302 |
| | Hinge | 0 | 0 | 15470 | 29413 |
| | SGA | 17 | 24149 | 34775 | 1206 |
| | SOAPdenovo2 | 5 | 9944 | 30565 | 1007 |
| | SPAdes | 7 | 7580 | 34733 | 1202 |

**Table 5** Assembly statistics of assemblers on *Bacillus cereus* dataset.

| Read type | Assemblers | # Contigs | N50 | L50 | GC(%) |
|-----------|-----------|-----------|-----|-----|-------|
| | A5-MiSeq | 180 | 94223 | 39 | 35.37 |
| | Canu | 68 | 858185 | 3 | 35.32 |
| **Long** | Falcon | 444 | 21016 | 87 | 35.45 |
| | Flye | 16 | 3779838 | 2 | 35.28 |
| | Hinge | 40 | 825555 | 4 | 35.31 |
| | SGA | 765 | 18848 | 75 | 35.47 |
| **Short** | SPAdes | 146 | 86886 | 17 | 35.34 |
| | SOAPdenovo2 | 535 | 24261 | 55 | 35.39 |

Notes.
  The minimum number of contigs generating 50% of the assembly base is represented by L50.

**Table 6** Assembly statistics of assemblers on *Caenorhabditis elegans* dataset.

| Read type | Assemblers | # Contigs | N50 | L50 | GC(%) |
|-----------|-----------|-----------|-----|-----|-------|
| | Canu | 102 | 5867748 | 1 | 62.67 |
| **Long** | Falcon | 857 | 15548 | 196 | 62.54 |
| | Flye | 1 | 5953794 | 1 | 62.71 |
| | Hinge | 52 | 3641048 | 2 | 62.47 |
| **Short** | SPAdes | 5 | 538 | 3 | 53.38 |
| | SOAPdenovo2 | 5 | 500 | 3 | 52.06 |

Notes.
  The minimum number of contigs generating 50% of the assembly base is represented by L50.

like misassemblies and mismatches. Consequently, in this genome, the assemblies with the largest contigs created the majority of the errors.

SOAPdenovo2, a short read assembler, generated a moderate N50 value of 216.1 kb while maintaining a short contig length. SPAdes, on the other hand, has the opposite problem. Both assemblers ranked first and second in terms of mismatch and indels when their mismatch statistics was examined. They were ranked fourth and fifth for recovering the *Escherichia coli* genome due to their excellent error handling.

**Table 7  Assembly statistics of assemblers on *Escherichia coli* dataset.**

| Read type | Assemblers | # Contigs | N50 | L50 | GC(%) |
|---|---|---|---|---|---|
| **Long** | A5-MiSeq | 173 | 197188 | 8 | 50.5 |
| | Canu | 94 | 4056254 | 1 | 50.55 |
| | Falcon | 1643 | 6143 | 431 | 49.78 |
| | Flye | 24 | 1072054 | 3 | 50.62 |
| | Hinge | 36 | 3356412 | 2 | 50.47 |
| | SGA | 114 | 196903 | 7 | 50.64 |
| **Short** | SPAdes | 413 | 196647 | 9 | 50.19 |
| | SOAPdenovo2 | 79 | 216146 | 6 | 50.68 |

Notes.
  The minimum number of contigs generating 50% of the assembly base is represented by L50.

**Table 8  Assembly statistics of assemblers on Human dataset.**

| Read type | Assemblers | # Contigs | N50 | L50 | GC(%) |
|---|---|---|---|---|---|
| **Long** | A5-MiSeq | 120438 | 602 | 47999 | 40.62 |
| | Canu | 5365 | 13580 | 1908 | 98.62 |
| | Falcon | 1961 | 14011 | 715 | 39.72 |
| | Flye | 1232 | 28044 | 381 | 40.39 |
| | HiFiasm | 15377 | 21918 | 6260 | 39.59 |
| | Hinge | 1591 | 14749 | 480 | 39.54 |
| | SGA | 917 | 905 | 140 | 47.12 |
| **Short** | SPAdes | 135 | 1882 | 30 | 50.94 |

Notes.
  The minimum number of contigs generating 50% of the assembly base is represented by L50.

Hinge, with an NGA50 of 72.9 kb, takes first place after the error within the contigs was corrected. SOAPdenovo2, A5-MiSeq, and SGA were next, with NGA values of 55.9 kb, 55.105 kb, and 55.1 kb, respectively. Hinge was unable to recover the genome as well as A5-MiSeq due to high structural error.

When the quality of the assemblers is compared to the amount of reference genome they retrieve, A5-MiSeq comes out on top with a genome fraction of 91.246%, followed by 91.188% by SGA and 91.082% by SPAdes. The genome fractions are listed in Table 3.

## Human genome

A5-MiSeq generated the highest contigs for the Human genome assembly, while Flye, HiFiasm, and Hinge produced the longest contigs. As compared to assemblies with the highest contigs, A5-MiSeq came in last with an N50 of 0.9 kb, followed by SGA with an N50 of 1.8 kb, and SPAdes with an N50 of 197 kb (Table 8).

The short-read assemblers were limited in their assembly due to the repetitive nature of the human genome and its size. When comparing the N50 value with the long-read assembler, SPAdes performed moderately, but it was shortlisted when comparing the longest contiguous reads. It has one of the best error management procedures for this genome.

**Table 9  Assembly statistics of assemblers on *Saccharomyces cerevisiae* dataset.**

| Read type | Assemblers | # Contigs | N50 | L50 | GC(%) |
|-----------|-----------|-----------|------|-----|-------|
| | A5-MiSeq | 369 | 91325 | 80 | 38.12 |
| | Canu | 193 | 710827 | 7 | 36.20 |
| **Long** | Falcon | 490 | 13483 | 309 | 27.58 |
| | Flye | 26 | 904913 | 8 | 38.19 |
| | Hinge | 110 | 754764 | 13 | 37.74 |
| **Short** | SPAdes | 376 | 93238 | 40 | 38.14 |
| | SOAPdenovo2 | 293 | 234358 | 17 | 38.13 |

Notes.
The minimum number of contigs generating 50% of the assembly base is represented by L50.

Flye, with an NA50 of 26.1 kb, takes first place after error correction. HiFiasm, Falcon, and Canu are the next three with NA50 values of 21.5 kb, 12.9 kb, and 10.2 kb, respectively. Flye was unable to recover the genome as good as HiFiasm due to high structural error. HiFiasm comes out on top with a 10.4% genome fraction, followed by A5-MiSeq with 2.437% and Canu with 1.383%. The genome fractions are listed in Table 3.

### *Saccharomyces cerevisiae*

Falcon performed the worst for *Saccharomyces cerevisiae*. Despite producing the highest contigs, it was only able to generate the longest contig, up to 23.2 kb out of a total length of 5.9Mb, resulting in the lowest N50, NA50, and genome fraction values.

SPAdes generated one of the longest contigs, with a moderate N50 value. SOAPdenovo2, on the other hand, produces a modest contig length while producing one of the highest N50 values. As compared to some of the long-read assemblers, both short-read assemblers performed well in minimizing mismatch and misassemblies (Table 4).

It can be seen from Table 9 that A5-MiSeq, Canu, and Flye outperformed the other assemblers in terms of N50 value. A5-MiSeq has the least mismatches and indels (63.7 kb and 5.7 kb), but its total number of uncalled bases (N's) was high in the assembly (6.1 kb).

Flye has 0 unaligned contigs according to the unaligned statistics. Since contigs contain structural errors, this does not always guarantee a 100% genome fraction. In comparison to A5-MiSeq and Flye, Flye has a high rate of misassembly errors (translocation and inversion). It does not retrieve the reference genome as good as Canu (Table 3), which has the largest genome fraction of 95.26%. Flye's ability to extract the reference genome to its full capacity is hampered by misassembly errors.

### *Staphylococcus aureus*

For *Staphylococcus aureus*, the highest number of contigs was achieved by Falcon, while Canu and Flye produced the largest contig and N50 value in the assembly as presented in Table 10. Based on statistics without reference, the performance of Canu was the best in terms of generating the highest N50 and N75 value.

Although SPAdes and SOAPdenovo2 produced modest contiguous reads, they faced a challenge in producing an N50 values that was above average. This was because they produced the most mismatches and indels (Table 4).

**Table 10  Assembly statistics of assemblers on *Staphylococcus aureus* dataset.**

| Read type | Assemblers | # Contigs | N50 | L50 | GC(%) |
|---|---|---|---|---|---|
| **Long** | A5-MiSeq | 36 | 244991 | 9 | 32.69 |
| | Canu | 9 | 2907970 | 1 | 32.73 |
| | Falcon | 2035 | 9071 | 746 | 33.21 |
| | Flye | 1 | 2896520 | 1 | 23.74 |
| | Hinge | 20 | 929468 | 3 | 32.73 |
| | SGA | 81 | 109236 | 17 | 32.66 |
| **Short** | SPAdes | 47 | 127345 | 8 | 32.66 |
| | SOAPdenovo2 | 41 | 180321 | 5 | 32.69 |

Notes.
  The minimum number of contigs generating 50% of the assembly base is represented by L50.

Canu encounters a 2.9 Mb of misassembled contigs in misassemblies while using a reference genome. Alternatively, the same issue exists with Flye's with a 2.8 Mb of misassembled contigs. Although it appears to be a significant challenge, they were able to overcome it by creating longer contigs that enabled them to maintain similar NG50 values to their N50 and higher NGA50 values. With the same defect, however, Hinge outperformed Canu and Flye, with a higher total aligned length value and an NGA50 value nearly twice as high as the other assemblers. As listed in Table 3, Hinge is the best assembler for *Staphylococcus aureus*, with a genome fraction of 92.5%.

# CONCLUSION

Considering the analyses of the seven genomes for which the actual assembly is available, Flye has consistently demonstrated superior output based on contig size, with the trade-off between scale and error rate. Hinge and Canu, though they demonstrate more errors than Flye, both performed reasonably well too. Falcon appears very capable of generating contigs, but its contigs contain several minor errors. Hinge and Canu, are based on the OLC method, with extra errors being added by both the assemblers to maximize the length of the contigs. Flye, by contrast, is a hybrid assembler with modules from other assemblers for many of its core functions, and its efficiency is not independent in this regard.

The short read assemblers SOAPdenovo2 and SPAdes provided results that were neither the best nor the worst, but on closer inspection revealed many errors that would not have been visible if the reference genome had not been present. Despite their average performances, they excelled at overcoming misassemblies. Except for a higher number of uncalled bases, a lower value of the largest contig, and a value of N50, SOAPdenovo2 showed an average rate of the number of contigs, replication ratio, indels, and misassembly for all sequenced genomes. We may conclude that the DBG approach is ideally suited for the handling of misassembly. Another thing we noticed about the short-read assembly is their low assembling time compared to the long read assemblies.

In the Human genome dataset, HiFiasm outperformed other assemblers in terms of genome fraction, total aligned length, and total length of contigs. It does, however, have its own set of problems with mismatches and misassemblies. On the human genome, it was

the only assembler that came close to Flye. Due to the limitation of PacBio HIFI reads, we were unable to assemble the rest of the dataset using HiFiasm.

In general, if the assembler's objective is to generate longer contigs, an assembler will generate a large N50 value, which would be costly in terms of errors. However, it would be at the cost of smaller contigs if the objective of an assembler is to minimize the number of errors. Larger N50 values were often created by Flye, Canu, and Hinge at the cost of accuracy (especially with Hinge).

Flye tends to be the most reliably performing assembler, in terms of contiguity as well as correctness, given all metrics.

### Funding
This research was supported by Basic Science Research Program through the National Research Foundation of Korea (NRF) funded by the Ministry of Education (NRF-2019R1F1A1064019). The funders had no role in study design, data collection and analysis, decision to publish, or preparation of the manuscript.

### Grant Disclosures
The following grant information was disclosed by the authors:
Ministry of Education: NRF-2019R1F1A1064019.

### Competing Interests
The authors declare there are no competing interests.

### Author Contributions
- Firaol Dida conceived and designed the experiments, performed the experiments, analyzed the data, performed the computation work, prepared figures and/or tables, authored or reviewed drafts of the paper, and approved the final draft.
- Gangman Yi conceived and designed the experiments, analyzed the data, authored or reviewed drafts of the paper, and approved the final draft.

### Data Availability
The data and code are available at GitHub: https://github.com/Firaol1221/Empirical-Evaluation-of-Methods-forDe-Novo-Genome-Assembly.

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
