# Peer review of "Empirical evaluation of methods for de novo genome assembly"

_PeerJ Computer Science, doi:10.7717/peerj-cs.636_

## Round 0.1 · original submission · Major Revisions

All three reviewers require major revisions. I believe that you may be able to reframe the topic to validate methods and improve the description of the experimental design to make a check of the validity adequate.

Reviewer 1 ·

Basic reporting

no comment

Experimental design

I do not see a real research question that the authors have investigated in this paper. They do not state how their work fills an identified knowledge gap - reassembly of well-known reference genomes is not a gap.

Validity of the findings

no comment

Additional comments

While generally well written and present, I fail to see what the authors were trying to study in this manuscript. Reframing the topic in the sense of validation of certain hypotheses about particular species (and including more than 1 genome) would certainly help.

Reviewer 2 ·

Basic reporting

The article is comprehensive and describes in detail most of the widely used assembly methods. The literature review is very thorough. The results are well described and convey the drawbacks and advantages of different assembly tools. One of the figures needs more work, but everything else looks good.

Experimental design

The experiments are well designed and the metrics on which assemblies are evaluated are correct and widely used in the literature. I have a couple of comments here. First, the data used for experiments are not very well defined - for eg, the coverage of data, the technology, error rate, etc. Second, one of the assemblers called HiFiAsm is not used for the evaluation. This assembler is currently the best in class so I would like to see it compared with the existing tools in the manuscript. Also, please provide the refseq/genebank accession for the reference genomes used for evaluation. Same for raw reads - please provide SRA accessions if the data was used from there. Although assemblers are compared across assembly metrics, the runtime/memory of different assemblers are not provided. For eg, Canu typically requires more memory and runtime compared to other assemblers. Please provide this information. Please also provide the metrics for one of the larger genomes such as the human genome.

Validity of the findings

1. Please make numbers right-aligned2. in the tables for better interpretation.
2. Figure 4 is confusing. Please make it as bar plot and color bars by the assembly algorithm.
3. Please provide the assemblies generated by different tools in a public bucket/link.

Reviewer 3 ·

Basic reporting

This manuscript reports on the state of genome assembly software and performs comparative testing on a number of such packages. The introduction does a decent job of laying out an overview of the current state of assembly. This section is followed by a "De novo assembly method" section which really seems to be part of the introduction in that it mostly describes major assembly algorithms in detail, but confusingly also seems to contain results figures. I can't find a true "Methods" section and this is a significant issue as it prevents me from understanding the Results that are being presented. Overall, I find this paper lands somewhere in-between a mini-review and a research article (but without a true Methods section) and doesn't live up to either label.

I find it difficult to make specific comments due to the lack of methods and unusual organization of the paper. I will note that I found the author's choices of organisms to use for testing a bit limited as it included three bacteria, a yeast, and a very small plant genome. The lack of metazoans or other larger eukaryotic organisms severely limits any ability to truly assess sequencer performance across organisms.

Experimental design

The experimental design is not described in any detail -- No Methods section is included. A brief mention in the results that default settings were used for each software is neither sufficient -- nor likely wise -- as default settings are not applicable in all circumstances. There is no indication of the source of data that is being used in the results section: importantly the details of what sequencing platform(s) each comes from, the average quality and depth of the data for each, etc is omitted completely. The study also seems to employ all assembly software against each dataset with no differentiation of whether a given package is optimized for long or short-read data -- which seems misleading to present results in this way.

Validity of the findings

Validity cannot be assessed as the experimental design is not adequately explained.

Additional comments

None.

---

## Round 0.2 · accepted · Accept

The reviewers have no further objections. They decided that the revised draft has improved, and deserves publication.

Reviewer 1 ·

Basic reporting

The revised article has improved, and I have no further qualms about it.

Experimental design

Methods are well described and appropriate

Validity of the findings

Findings are well supported by the literature and data

Additional comments

I appreciate the revised manuscript.